# Plant-Produced Viral Nanoparticles as a Functionalized Catalytic Support for Metabolic Engineering

**DOI:** 10.3390/plants13040503

**Published:** 2024-02-11

**Authors:** Christian Sator, Chiara Lico, Elisa Pannucci, Luca Marchetti, Selene Baschieri, Heribert Warzecha, Luca Santi

**Affiliations:** 1Plant Biotechnology and Metabolic Engineering, Technical University of Darmstadt, Schnittspahnstrasse 4, 65287 Darmstadt, Germany; 2Centre for Synthetic Biology, Technical University of Darmstadt, Schnittspahnstrasse 4, 65287 Darmstadt, Germany; 3Laboratory of Biotechnologies, Italian National Agency for New Technologies, Energy and Sustainable Economic Development, ENEA, Casaccia Research Center, Via Anguillarese 301, 00123 Rome, Italy; chiara.lico@enea.it (C.L.); selene.baschieri@enea.it (S.B.); 4Department of Agriculture and Forest Sciences (DAFNE), University of Tuscia, Via S. Camillo De Lellis, 01100 Viterbo, Italy; e.pannucci@unitus.it (E.P.); luca.marchetti@unitus.it (L.M.); luca.santi@unitus.it (L.S.); 5Laboratory of Biomedical Technologies, Italian National Agency for New Technologies, Energy and Sustainable Economic Development, ENEA Casaccia Research Center, Via Anguillarese 301, 00123 Rome, Italy

**Keywords:** plant metabolic engineering, plant virus nanoparticles, plant-produced virus-like particles, multi-enzymatic assemblies, olivetolic acid, cannabinoids

## Abstract

Substrate channeling could be very useful for plant metabolic engineering; hence, we propose that functionalized supramolecular self-assembly scaffolds can act as enzymatic hubs able to perform reactions in close contiguity. Virus nanoparticles (VNPs) offer an opportunity in this context, and we present a functionalization strategy to display different enzymes on the outer surface of three different VNPs produced in plants. Tomato bushy stunt virus (TBSV) and Potato virus X (PVX) plant viruses were functionalized by the genetic fusion of the E-coil peptide coding sequence to their respective coat proteins genes, while the enzyme lichenase was tagged with the K-coil peptide. Immobilized E-coil VNPs were able to interact *in vitro* with the plant-produced functionalized lichenase, and catalysis was demonstrated by employing a lichenase assay. To prove this concept *in planta*, the Hepatitis B core (HBc) virus-like particles (VLPs) were similarly functionalized by genetic fusion with the E-coil sequence, while acyl-activating enzyme 1, olivetolic acid synthase, and olivetolic acid cyclase enzymes were tagged with the K-coil. The transient co-expression of the K-coil-enzymes together with E-coil-VLPs allowed the establishment of the heterologous cannabinoid precursor biosynthetic pathway. Noteworthy, a significantly higher yield of olivetolic acid glucoside was achieved when the scaffold E-coil-VLPs were employed.

## 1. Introduction

Metabolic engineering has become a major field in modern biotechnology as it promises the deliberate construction of enzymatic pathways leading to the formation of any given natural product or—even more—to new-to-nature compounds. Since more and more natural pathways are deconstructed on an enzymatic and genetic level, it has been shown that many catalyst sequences can be rebuilt in heterologous hosts. Detaching the biosynthesis from its natural producers promises a more focused production of target molecules with fewer side products, an increase in exploitable product levels, and also a more sustainable production. Plant natural products are especially of great interest for engineering efforts since they represent a vast diversity of complex chemical structures as well as various uses such as fine chemicals, nutrients, and pharmaceuticals. When it comes to heterologous production, most effort so far has gone into the engineering of fermentable microorganisms. Although *Escherichia coli* and other bacteria are easy to manipulate and can be readily fermented in standard lab equipment with scale-up options, they are only of limited use for the establishment of complex plant pathways. As bacteria usually lack membranous cellular compartments, they are poor hosts for complex pathways involving membrane-anchored catalysts like P450 monooxygenases. Hence, the eukaryotic *Saccharomyces cerevisiae* became the working horse for the heterologous establishment of plant metabolic pathways. This is exemplified by many successful implementations of heterologous pathways leading to the formation of plant natural products in yeast [1]. However, in most cases, the obtainable level of metabolites lags well behind what can be achieved by the natural producer, like cannabinoids [2] or tropane alkaloids [3]. This frequently made observation suggests that a pathway not only comprises a series of enzymes but also requires a context of precursor supply, spatiotemporal regulation and distribution over a series of cell types and compartments, as well as flux control.

As plants provide all the requirements, it is comprehensible that they are considered as a potential production platform. This is especially true since transient transformation has been established as a fast and scalable method for plant production with no need for the laborious and time-consuming generation of transgenic plant lines.

Metabolic channeling is a naturally occurring process that consists of a series of enzymatic reactions taking place in such proximity that the intermediates are less likely released to the bulk phase but readily available for the subsequent catalytic steps. In this context, particularly intriguing is the possibility of exploiting substrate channeling to maximize the efficiency of enzymatic pathways [4,5]. Thus, multi-enzymatic clustering bears the potential to direct the metabolic flux, improving the overall biosynthetic performance [6]. For this purpose, inorganic and organic frameworks have been explored, and nano-scaffold architectures displaying different enzymes on a single entity, particularly those based on DNA, peptides, proteins, liposomes, and nanoparticles, have attracted widespread attention [7,8]. However, the precise organization of these complex nanostructures is not trivial for triggering the proper substrate channeling effect *in vitro* and especially in vivo [9].

In this respect, plant virus nanoparticles (VNPs) and in planta assembled virus-like particles (VLPs) may represent ideal nano-scaffold candidates since they offer different possible shapes, sizes, chemical reactivities, structural stabilities, dynamic properties and noteworthy their production is based on environmentally friendly processes. Most importantly, their outer surface is highly programmable, making tailored structural engineering for multi-enzyme display an achievable task.

Thanks to these characteristics and to the improved immunogenic, biocompatibility, and biodegradability properties, VNPs are already used in several different fields of application, especially in nanotechnology, for example, to produce bio-batteries and in nanomedicine, for the development of new vaccine formulations, diagnostic systems, immunotherapy, and targeted drug delivery platforms [10,11].

The functionalization of the VNP surface, for example, for the display of heterologous proteins, such as enzymes, can be obtained by fusing the desired coding sequences to the viral coat protein (CP) gene by genetic engineering or attached by chemical modification to exposed CP amino acids using different available bioconjugation strategies after VNP assembly has occurred [12]. Alternatively, as explored in this study, complex heterogeneous supramolecular structures can be obtained via the non-covalent modification of VNPs using natural or de novo designed affinities of proteins, or protein domains, that will function as molecular bridges when attached to the different enzymes and to the VNP CPs, respectively [13]. In this regard, a particularly compelling example is the heterodimerizing peptides such as those derived from alpha-helical coiled-coil domains. These peptides interact specifically and form stable heterodimeric coiled-coil structures, which have diverse functions in nature, playing crucial roles in various cellular processes such as providing stability to protein-derived fibers and filaments [14] and to the assembly structures of molecular motors [15]. Moreover, they are part of transcription factors and signaling complexes, thus exerting a role in gene regulation and signal transduction, respectively [16,17]. In particular, the E-coil structure is based on several repetitions of the negatively charged glutamate, while the K-coil of the positively charged lysine. The electrostatic interactions within the coiled-coil structure may provide a mechanism for molecular recognition, allowing the peptides to specifically target and interact with complementary partners [18,19]. This feature has been successfully applied in nanobiotechnology using different heterologous expression systems for the most diverse applications, ranging from drug delivery to diagnostic and material science [20].

The aim of the present study is to demonstrate that this strategy allows the generation of plant VNP—and VLP—multi-enzymatic assemblies by proving the catalytic performance *in vitro* and in planta, respectively.

To do so, a robust and reliable reporter system was needed to evaluate the *in vitro* binding capabilities of the VNPs, which we identified in the thermostable lichenase (licBM3) from *Clostridium thermocellum* [21]. LicBM3 shows hydrolytic capacity targeting specifically β-1,3-1,4-glucans [22]. This specificity allows the utilization of licBM3 as a reporter protein because it selectively hydrolyzes the substrate lichenane, and the resulting sugar moieties can be quantified to gain insight into the functionality and expression levels of the reporter [23]. Both enzymatic activity as well as substrate are virtually absent from plants, therefore omitting eventual background activity.

For the proof-of-principle, Tomato bushy stunt virus (TBSV) and Potato virus X (PVX) were used as nano-scaffolds displaying the E-coil peptide, while the lichenase enzyme was tagged with the K-coil peptide. TBSV and PVX have totally different structures and dimensions. TBSV is an icosahedral virus of about 30 nm in diameter composed of 180 identical CP units embedding a single-stranded positive sense (ss(+)) RNA genome [24]. PVX has a filamentous flexible structure about 500 nm in length and 13 nm in diameter, composed of approximately 1300 CP units, in this case also embedding an ss(+) RNA molecule [24,25]. Both viruses are ideal for establishing highly ordered, multivalent scaffolds for enzyme interaction platforms.

To test VNP scaffolds in vivo, both PVX and TBSV are not ideal due to the fact that it is difficult to combine and synchronize the infection (for TBSV-E-coil and E-coil-PVX) and the agroinfiltration (for the K-coil enzyme) procedures. Therefore, we also included the nucleocapsid of the hepatitis B virus (HBV) formed by the HBV core antigen (HBcAg) in our studies. HBc-based VLPs are indeed one of the first, best characterized, and widely employed VLPs. The HBc core protein forms nanometric icosahedral VLPs that, over the years, have been produced in virtually all major expression systems, including plants, by expressing the corresponding gene [26]. Moreover, the fine structure of HBc VLPs and their monomers has been elucidated in detail at the crystal level, and this has led to the detection of multiple sites for the insertion of peptides aiming at different types of functionalization, with epitope display being the most relevant. A truncated form of the core antigen has been proven sufficient to form particles of 30 or 34 nm in diameter, depending on the number of dimers forming the particles (90 or 120, respectively) ([27,28]). More important and other than TBSV and PVX, HBcAg does not need to incorporate nucleic acids for VNP formation, and heterologous molecules can be attached at three different locations within the core protein, at both N- and C-terminal part of the core protein, protruding to the virion surface, and into the so-called MIR region [27]. As a proof of concept to test the performance of the system with a relevant enzymatic pathway, three enzymes were chosen that subsequently convert hexanoic acid into olivetolic acid, the key intermediate in the biosynthesis of phytocannabinoids. Here, the Hepatitis B core (HBc) VLPs, similarly functionalized with the E-coil peptide, were tested in *Nicotiana benthamiana* plants via the co-expression of K-coil-modified pathway enzymes. Previous studies have shown the formation of olivetolic acid from hexanoic acid via the heterologous expression of the three pathway enzymes acyl-activating enzyme 1, olivetol synthase, and olivetolic acid cyclase. However, further downstream metabolization toward cannabinoids seems to be hindered by endogenous glycosyltransferases from *N. benthamiana* or due to possible toxic effects of intermediates [29]. These are both hurdles that might be overcome by the utilization of VLPs for metabolic channeling of this biosynthetic pathway.

## 2. Results and Discussion

### 2.1. In Planta Production of TBSV-E-coil Nanoparticles, E-coil-PVX Nanoparticles, and K-coil-Lichenase

Isolated structural motives involved in protein dimerization have been extensively and successfully used *in vitro* for different applications, such as developing affinity reagents and tags for the detection, purification, and characterization of recombinantly expressed peptides and proteins. For these properties, they have been sometimes referred to as “velcro” peptides, and their employment has been more recently extended to in vivo applications to bring proteins together inside a cell and, lately, to functionalize NPs [30,31,32]. They are particularly relevant as far as VNPs and plant derived VNPs are concerned. In fact, although VNPs offer a great plasticity in terms of functionalization due to the possibility, besides other strategies, of genetically engineering the respective CPs, they might suffer from constraints related to the dimension of proteins that can be attached and displayed. This is mainly attributable to steric hindrances preventing proper self-assembly, and, in this context, heterodimerization peptides are of great value, providing a non-covalent strategy to functionalize CPs *in vitro* or also in vivo after self-assembly has occurred. In particular, we have chosen the E-coil (EVSALEK) and K-coil (KVSALKE) peptides originally developed de novo by Tripet and coauthors [18,19]. They are based on alpha-helical coiled-coil, which are widespread structural domains with a variety of functions, among which heterodimerization is one of the majors. Each individual helix contains several heptad repeats (four in the present study) that mediate electrostatic interactions between the two helices of opposite charge.

TBSV-E-coil NPs were generated by the insertion of the sequence encoding the E-coil peptide (striped box, Figure 1a) into the TBSV–vector at the 3′-end of the p41 gene, corresponding to the C-terminus of the CP, known to be exposed on external virion surface [33,34] (Figure 1a). Instead, E-coil-PVX was obtained by genetic fusion between the E-coil peptide (striped box, Figure 1b) and the viral CP at the 5′-end of the *cp* gene, as in this case, the N-terminal part of the protein is exposed on the surface of the virion [35]. Since PVX has some constraints for the display of long and complex proteins [36], we spaced the E-coil sequence from the PVX CP through the 16-amino acid long 2A sequence derived from the foot-and-mouth disease virus (FMDV) [37,38] (grey box, Figure 1b). This peptide confers a cotranslational ribosomal skip, leading to the formation of mosaic particles consisting of WT and E-coil fused CPs in a 9:1 fixed ratio. In this way, the presence of the heterologous sequence does not hamper CP folding or virion shell assembly since it is not fused to all viral CPs [25,39]. *N. benthamiana* plants have been inoculated with *in vitro* transcribed TBSV genome or with the expression plasmid encoding the whole PVX genome. Both WT and engineered viruses behaved similarly in timing of symptoms onset and phenotypic appearance on both inoculated and systemic leaves (Figure 1c,d). The analysis of the genetic stability of the chimeric viruses via several re-infection cycles confirmed that the genetic modifications did not alter the assembly and fitness of both TBSV and PVX. All the VNPs were purified by homogenizing the infected tissues in an extraction buffer (for TBSV under acidic conditions), followed by clarification, precipitation, ultracentrifugation, and sucrose cushion (see details in the Material and Methods Section). Purified particles were then analyzed using SDS-PAGE. For TBSV, silver staining proved the purity of the NP batch and showed bands corresponding to WT CP or CP-E-coil at the expected molecular sizes (Figure 1e: 41 kDa, black arrow, and 45 kDa, grey arrow, respectively). The final yield of VNPs, quantified by absorbance measurements and by using the TBSV extinction coefficient, was 794 μg/g for TBSV-wt and 810 μg/g for TBSV-E-coil. Purified PVX particles separated on SDS-PAGE showed bands corresponding to mosaic NPs composed of non-fused and fused CPs (Figure 1g: 25 kDa, black arrow, and 30 kDa, grey arrow, respectively), in line with the expected ratio previously reported, together with additional bands corresponding to multimers and degradation products (Figure 1g). The final yield, quantified as before by absorbance measurements and by using the PVX extinction coefficient, was 250 μg/g and 92 μg/g for PVX-wt and E-coil-PVX, respectively. The definitive proof of proper assembly and morphology of TBSV-E-coil and E-coil-PVX were assessed by TEM, and indeed, both resulting chimeric VNPs were identical to their WT counterparts (Figure 1f,h).

To evaluate the interaction properties with previously described TBSV-E-coil and E-coil-PVX, the thermostable lichenase from *Clostridium thermocellum* (licBM3) was used. Fusion constructs of licBM3 functionalized with the corresponding peptide linker were generated in order to analyze the binding capabilities of the nano-scaffolds. In the experimental setup, we also introduced, as a negative control, the fusion construct GFP::licBM3 (GenBank, KX458181), already available as previously used by the group [23], where the lichenase is fused to the GFP, retaining its glucosidase activity, but in the absence of the K-coil peptide.

The transcriptional units for the production of lichenase fused with either K-coil (LiBM3:K-coil) or GFP (LiBM3:GFP) were transiently expressed in *N. benthamiana* leaves, as previously reported ([23]; Appendix A). The total soluble protein extracts were obtained from the infiltrated leaves, and the protein extracts were used for *in vitro* binding assays.

### 2.2. E-coil Plant VNPs Interact with K-coil-Lichenase In Vitro

A Sandwich-ELISA protocol, with E-coil plant VNPs immobilized on ELISA plates, ready to capture the enzyme present in the extract of plants agroinfiltrated with the K-coil–Lichenase construct, was set up and utilized to evaluate the interaction properties of the coiled-coil peptide pair. As a control, the extract of a non-treated plant and the extract of a plant agroinfiltrated with the GFP–Lichenase were also used to verify that the virus–enzyme interaction was specifically ascribable to the E-coil/K-coil interaction. After incubation and washing, the lichenan substrate was supplied to measure lichenase activity. The test, quick and straightforward, is based on the quantification of free-reducing sugar moieties released due to substrate hydrolysis after their reaction with the dinitrosalicylic acid (DNS) reagent [40,41].

Data clearly indicate that the K-coil enzyme is able to specifically bind to engineered E-coil VNPs, both spherical (TBSV) and filamentous (PVX), maintaining its catalytic activity (Figure 2). Moreover, in the case of PVX, it has come to light that to have the appropriate interaction with K-coil–Lichenase it is necessary to coat the plates with 4 times more PVX particles compared to TBSV. The result was indeed expected since the use of the 2A peptide to generate mosaic nanoparticles, necessary to support proper virion assembly, drastically reduced the density of the E-coil peptide on the PVX virion surface, resulting in particles with only 10% of chimeric CPs displaying the E-coil peptide required for the interaction.

These data clearly support the use of E-coil and K-coil peptides to functionalize the outer surface of pVNPs with large enzymatic proteins that, in the described case, are also able to retain proper catalytic activity. This indeed set the ground for their *in planta* use.

### 2.3. Production of HBc VLPs in Plants

The feasibility of E-coil and K-coil to directly mediate interactions in planta was addressed by employing multiple enzymes that were simultaneously expressed and tested together with VNPs to obtain the proof of concept that these high-order self-assembly NPs and enzyme clusters would act as a catalytic hub to foster metabolic channeling. Since for this purpose, we did not consider full infectious viruses, as those previously described, optimal mainly for issues related to synchronization of enzymes’ expression provided by agroinfiltration and VNP expression obtained by infection, we decided to use the HBc-based VLPs. In fact, similarly to the enzyme’s counterparts, expression can be achieved by agroinfiltration, in this case, using the gene encoding for HBV core protein.

HBc-based VLPs offer a multitude of possible functionalization sites. Among these sites, the MIR region (127–133) is exposed on the VLPs surface, forming a small protrusion [27]. For this reason, this region is often used to fuse and expose the protein/fragment of interest. Nonetheless, the E-coil peptide has a particular structure and must be able to interact with its K-coil counterpart, so anchoring both ends did not seem the appropriate choice. On the other hand, in addition to the MIR, also the end of the C-terminal helix is accessible and forms small protrusions on the surface of the HBc particle [27]. For this reason, the E-coil peptide has been fused to the C-terminus region of the truncated HBc protein (aa 1–150) via the GoldenBraid system (Figure 3a). After *A. tumefaciens* transformation and plants agroinfiltration, plant extracts of total soluble proteins were analyzed by Coomassie-stained SDS-PAGE, highlighting some differential bands between plants agroinfiltrated with HBc constructs and agroinfiltrated control plants (only p19 construct), even if slightly shifted in comparison to what expected. Yet, distinct protein bands were observed between marker bands for 25 and 35 kDa, similar to what has been already reported in the literature [42,43] (Figure 3b), with the higher band presumably corresponding to a dimer, which indeed acts as a pivotal intermediate in the assembly of core particles [44], or to some additional high molecular weight quaternary structures which may represent further intermediate aggregation states. A purification protocol was set up based on differential centrifugation and on a sucrose cushion. The quality of the purified batch was verified via a Coomassie-stained SDS-PAGE (Figure 3c, arrow indicates the monomeric purified HBc subunit), and the final yield was calculated to be 15 μg/g for HBc-wt and 10 μg/g for HBc-E-coil. Finally, the morphology and dimension of purified VLPs were assessed by TEM (Figure 3d). This analysis confirmed that HBc were able to correctly self-assemble in plant cells forming VLPs of about 30 nm in diameter, similar to native HBV nucleocapsid, confirming previous results indicating that particles generated by truncated HBc have a slightly larger diameter (30 nm) than the full-length particles (28 nm) [45,46].

### 2.4. HBc VLPs Can Serve as a Hub for Multi-Member Enzymatic Cascades in Planta

#### Co-Expression in Planta of HBc-E-coil and AAE1-, OLS-, and OAC-K-coil

In order to evaluate the beneficial effect of a molecular assembly on a given metabolic pathway, we chose a three-enzyme sequence from the cannabinoid pathway yielding the intermediate olivetolic acid from the precursor hexanoic acid. The three enzymes comprising the biosynthetic pathway for olivetolic acid production (acyl-activating enzyme 1, AAE1; olivetol synthase, OLS; and olivetolic acid Cyclase, OAC) (Appendix A) are soluble, cytosolic enzymes that do not require complex cofactors. Additionally, the product olivetolic acid rapidly undergoes glycosylation by endogenous *N. benthamiana* glycosyl transferases, forming a stable and easily detectable product and making the enzymes valid proof for VNP assembly and functionality. Therefore, the enzymes were modified with the coiled-coil linker pair in order to test the ability of the HBc VLPs to function as an enzymatic hub for metabolic channeling (Appendix A). The pathway enzymes were C-terminally fused with the K-coil sequence and transiently co-expressed with HBc particles, displaying the E-coil peptide in *N. benthamiana* plants. The HPLC-MS analysis of metabolites extracted from infiltrated leaves revealed the presence of a new metabolite peak corresponding to the *m*/*z* ratio of olivetolic acid glucoside (385.2 in negative SIM) at 6.7 min for the plants infiltrated with the transcriptional units for the production of the K-coil tagged pathway enzymes (AAE1:K-coil, OLS:K-coil, and OAC:K-coil) compared to *N. benthamiana* wild type (Figure 4A,B). The presence of this metabolite is a clear indication that the enzymes fused with the c-terminal K-coil sequence are functional and constitute the synthetic pathway for olivetolic acid production in *N. benthamiana* according to Geißler and coauthors, 2021 [47]. The co-expression of the modified pathway enzymes together with the scaffold protein (HBc-E-coil) led to noticeably higher peak intensities for the product peak (Figure 4C).

Using the area underneath the product peak as a metric to calculate the product yield revealed a significant increase in produced olivetolic acid glucoside in the plants expressing the functionalized enzymes together with the scaffolding NPs compared to plants expressing the enzymes but lacking the scaffold. Interestingly, in support of our hypothesis, the presence of the HBc NPs displaying the E-coil sequence led to a 4-fold increase in product formation (Figure 5).

## 3. Materials and Methods

### 3.1. TBSV and PVX Construct Design

Vectors carrying the full-length cDNA copies of the ss(+) RNA genomes of TBSV (pTBSV-p and pTBSV-v) [24] and PVX (pPVX201 and pPVX2A) [48] were used to clone the modified TBSV and PVX constructs (TBSV-E-coil and E-coil-PVX). For TBSV, the E-coil sequence was obtained by annealing a pair of sense and anti-sense oligonucleotides designed to generate a fragment with 5′ *ApaI*-compatible and 3′ *PacI*-compatible ends (Appendix A) to be ligated into properly digested pTBSV-v plasmid, generating pTBSV-E-coil (Appendix A). For the production of pPVX-E-coil (Figure 1b), a pair of sense and anti-sense oligonucleotides (Appendix A) designed to generate a fragment with 5′ NotI-compatible and 3′ *SalI*-compatible ends were *in vitro* annealed and ligated into properly digested pPVX2A plasmid (kindly provided by Prof. Gianpiero Marconi, University of Perugia, Italy). Oligonucleotides have been designed to respect the codon usage of *N. benthamiana*.

### 3.2. Production and Purification of TBSV and PVX Nanoparticles

The TBSV plasmids (encoding for wt and E-coil modified particles) were linearized with XmaI and *in vitro* transcribed using the MEGAscript T7 High Yield Transcription kit (Ambion, Lifetechnologies, Carlsbad, CA, USA), following the instructions of the manufacturer. The PVX plasmids were directly used to induce the infection of plants as the transcription of the cDNA encoding the whole viral genome is under the control of the 35S promoter of Cauliflower Mosaic Virus (CaMV), constitutively active in plant cells. *N. benthamiana* plants were grown in controlled conditions (16/8 h day/night cycle, 25 °C, 65% humidity, daily light integral 3.9 moles/day, photosynthetically active radiation 136 μmol/m^2^/s). At the age of 6 to 8 weeks, the adaxial surface of two leaves/plant was abraded with carborundum (Silicon carbide, VWR International, Radnor, PA, USA) and inoculated with (i) 20 μg/leaf of pPVX201 or E-coil-pPVX; or (ii) approximately 2 μg/leaf of TBSV-wt or TBSV-E-coil RNA. After symptoms appeared on systemic leaves (Figure 1c,d), the genetic stability of the modified viruses via re-infection cycles was assessed using RNA extraction, RT-PCR analysis, and sequencing, as previously reported [24,49]. Virus propagation and pVNP purifications were performed as already described ([24,50], avoiding the sucrose gradient step for PVX), and NPs were quantified by absorbance measurements at 260 nm and by using the TBSV and PVX extinction coefficients (4.5 and 2.97, respectively). The quality of the purification batches of both viruses was evaluated by separating NPs via SDS-PAGE with subsequent silver staining (Figure 1e,g).

### 3.3. GoldenBraid for HBc VLPs and Enzymes Production (K-coil Lichenase, Lichenase-GFP, AAE1-, OLS-, and OAC-K-coil)

Several transcriptional units for the in vivo production of the modified Hepatitis B nanoparticles in *N. benthamiana* plants were generated using the GoldenBraid cloning system [51]. All components for the transcriptional units were cloned in the Universal Domesticator plasmid (pUD). The domesticated sequences were afterward assembled into the destination vector pDGB3alpha1 according to the GoldenBraid protocol. Therefore, the domesticated plasmids for the coding regions were combined with the domesticated regulatory elements into complete transcriptional units in a one-pot reaction using BsmBI and T4-Ligase. The 35S-Promoter of Cauliflower Mosaic Virus (35S) and the terminator of nopaline synthase from *A. tumefaciens* (NOS) were used as regulatory elements for all cloned transcriptional units. Two transcriptional units for the production of HBc nanoparticles were generated: one for the production of wild-type Hepatitis B core protein (HBc-WT) and one for the production of HBc fused with the E-coil sequence at the c-terminus (HBc-E-coil). The domesticated coding sequence of HBc consists of the 5′-module(Residues 1–75), the major immunodominant region (MIR, residues 76–80), and the 3′-module coding for residue 81–149. The sequence for the 3′-part was truncated to remove the RNA binding domain.

Furthermore, several transcriptional units for the production of reporter proteins for *in vitro* binding assays were generated. Therefore, the K-coil sequence was fused to the N-terminus of the coding sequence for the reporter protein lichenase (K-coil-LicBM3) (Appendix A). The transcriptional unit was assembled inside the pGB3alpha1 destination vector utilizing domesticated pUDs containing the coding sequences for the K-coil sequence, the lichenase, and the regulatory elements (35S and NOS). A protein fusion from Lichenase and GFP was used as an additional reporter construct (LicBM3:GFP) (Appendix A).

Additional transcriptional units for the production of catalytic enzymes fused with the K-coil binding peptide were cloned to evaluate the potential of the designed nanoparticles to improve catalytic yields of heterologous biosynthetic pathways in planta. The K-coil sequence was fused to the C-terminus of the catalytic enzymes for the biosynthetic olivetolic acid pathway. The transcriptional units for the production of the tagged enzymes were cloned via GoldenBraid cloning. For the assembly reaction, pUDs containing the sequences for the catalytic enzymes (AAE1, OLS, and OAC) together with pUDs containing the regulatory elements (35S and NOS) and the K-coil sequence were used to clone the transcriptional units into the GoldenBraid destination vector (pGB3alpha1).

### 3.4. Agroinfiltration Procedure

For agroinfiltration experiments, all the assembled transcriptional units were transformed into chemically competent *A. tumefaciens* cells of strain EHA105 via heat shock.

The heterologous proteins were transiently expressed in plant tissues by agroinfiltration [52]. Briefly, *A. tumefaciens* EHA105 cells carrying the various constructs (HBc-wt, HBc-E-coil, Lichenase-K-coil, Lichenase-GFP, AAE1-, OLS-, and OAC-K-coil) were grown at 28 °C in YEB medium, added with Rifampicin (50 mg/L) and Kanamycin (50 mg/L), centrifuged (5000× *g* for 15 min), and the pellet was resuspended in infiltration buffer [10 mM 2-(N-morpholino) ethanesulphonic acid (MES), 10 mM MgCl_2_, 100 µM 3,5-Dimethoxy-4-hydroxyacetophenone (acetosyringon) pH 5.5]. The bacteria suspensions were used to infiltrate *N. benthamiana* leaves by submerging plants at 40 days from germination in a beaker containing the transformed bacterial suspension, and a 10 mmHg vacuum was then applied for 1 min in a vacuum cabinet. To enhance transient expression efficiency, plants were co-infiltrated with *A. tumefaciens* C58C1 (pCH32) cells carrying the binary vector p35S:p19, encoding the silencing suppressor protein P19 of TBSV (P19-TBSV) [53]. Agrobacteria strains were mixed at an optical density (O.D. 600 nm) of 0.5. All the leaves, except the apical ones (previously demonstrated to have a lower expression efficiency, [54]), were sampled at 5 days post-infiltration (d.p.i), immediately weighed, frozen in liquid nitrogen, and stored at −80 °C.

For what concerns the transcriptional units for the olivetolic acid pathway, they were co-infiltrated together with plasmids for the expression of the HBc-E-coil scaffold and the gene silencing suppressor p19, as well.

Four d.p.i. leaves were infiltrated with 4 mM hexanoic acid using a syringe without a needle. The day after, the infiltrated leaves were harvested, flash-frozen in liquid nitrogen, and stored at −20 °C.

### 3.5. HBc VLPs Purification

HBc-E-coil and wt constructs were inserted in the *A. tumefaciens* EHA105 strain and used to perform plant agroinfiltration, as described above. Agroinfiltrated tissue was used to perform a total soluble protein extraction, as already described, and the extracts were quantified via a Bradford assay, normalized, and run on an SDS-PAGE, with subsequent final Coomassie staining. VLPs were purified (following a protocol modified from [43,55,56]) by extracting the agroinfiltrated tissue 1:3 *w/v* in PBS, 1 mM EDTA, pH 5.2, Triton X-100 0.05%. After filtration through the Miracloth membrane and incubation for 1 h on ice, the sample was centrifuged at 15,000× *g* 30 min, and the pH of the recovered supernatant was adjusted to 7.0. After incubation for 1 h on ice and centrifugation 8000× *g* 30 min, the sample was loaded on a 20% sucrose cushion prepared in PBS, 1 mM EDTA, pH 7.0, and centrifuged 140,000× *g* 2 h. The pellet was resuspended in PBS, 1 mM EDTA, pH 7.0, loaded on an SDS-PAGE and Coomassie-stained, and then used to characterize VLPs by TEM.

### 3.6. Transmission Electron Microscopy

For TEM images, samples were fixed using 4% paraformaldehyde (PFA) in PBS, placed on formvar-carbon-coated copper grids, washed, and negatively stained with 2% uranyl acetate. Samples were observed using a JEOL 1200 EX II electron microscope (JEOL Ltd., Akishima, Tokyo, Japan), and images were acquired using the Olympus SIS VELETA CCD camera (Olympus, Münster, Germany) equipped with the iTEM software version, iTEM 5.1. To calculate virion diameter, micrographs were analyzed using the scientific image manipulation software ImageJ, version 1.54r (National Institutes of Health, Bethesda, MD, USA). TEM analysis was performed by the Electron Microscopy Lab, Centro Grandi Attrezzature (CGA), University of Tuscia, Viterbo.

### 3.7. Sandwich-ELISA Assay

The interaction of the Lichenase-K-coil with viral E-coil-NPs and its activity was assessed *in vitro* by an assay developed in a multiwell plate. Overall, 4 μg of PVX or 1 μg of TBSV NPs diluted in 1× phosphate-buffered saline (PBS: 151 mM NaCl, 8.4 mM Na_2_HPO_4_·12H_2_O, 1.86 mM NaH_2_PO_4_·H_2_O, pH 7.2) were distributed in triplicate into the wells of an ELISA plate and incubated O.N. at 4 °C. After blocking for 2 h at 37 °C in 5% non-fat milk solution prepared in PBS, plates were washed 3 times with PBST (PBST: PBS, 0.05% TWEEN-20) and twice with PBS. Then, agroinfiltrated tissue was used to perform a total soluble protein extraction by grinding the sample in PBS 1:3 *w*:*v*, followed by centrifugation at 20,000× *g* for 5 min, and 100 μL of the extract of plants agroinfiltrated with Lichenase–K-coil/Lichenase–GFP or *N. benthamiana* plants were added and incubated 2 h at 37 °C. After washing, as described before, the incubation with plant extracts was repeated. After washing, 100 μL of 1% lichenan (dissolved in H_2_O) were added and incubated at 70 °C for 3 h. The content of each well was transferred into a 1,5 mL Eppendorf, and then 400 μL of H_2_O, 500 μL of 3, 5-dinitrosalicylic acid (DNS) reagent, and 165 μL 40% K-Na-tartrate were added. After incubation at 95 °C for 10 min, samples were put on ice for 15 min and transferred into a cuvette to measure absorbance at 540 nm.

### 3.8. HPLC/MS Analysis of Metabolic Channeling Effect of Modified HBc VLPs Functioning as Metabolic Hubs for Biosynthetic Cannabinoid Pathway

For the extraction of plant metabolites, 200 mg of frozen plant matter was ground up. Then, 400 µL methanol (80%) was added, and the plant matter was broken up in an ultrasonic bath for 30 min. Afterward, the leaf debris was removed by 2 centrifugation steps (17,000× *g* 10 min), and the clear supernatant was moved to HPLC vials for analysis. Metabolite analyses were performed via HPLC-MS (Agilent Infinity 1260, Agilent Technologies, Santa Clara, CA, USA) with a reverse phase column (Poroshell 120 SB-C18 3.0 × 150 mm, 2.7 µm, Agilent Technologies, Santa Clara, CA, USA) as stationary phase. Acetonitril was used for the mobile phase and the phase composition is listed in Table 1.

The produced olivetolic acid (glucoside) was detected according to the *m*/*z* ratio (385.2) in negative single ion mode (SIM).

## 4. Conclusions

Our study shows that various virus nanoparticles can be functionalized with small, versatile heterodimerizing peptides, enabling the decoration with functional enzymes, both in vivo and *in vitro*. The formation of supramolecular complexes displaying functional metabolons can now be easily achieved due to its modular built, another step toward a synthetic biology approach for plant metabolic engineering.

## Figures and Tables

**Figure 1 plants-13-00503-f001:**
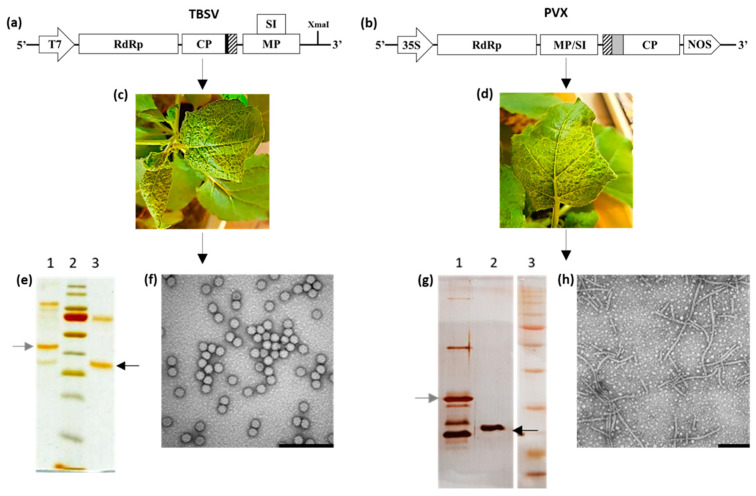
Production of TBSV-E-coil, TBSV-wt, and E-coil-PVX, PVX-wt NPs in plants. (**a**,**b**) Schematic diagram of genetic constructs: T7, T7 promoter; RdRp, RNA-dependent RNA polymerase; CP, coat protein; MP, movement protein; SI, silencing inhibitor; black box, linker; striped box, E-coil sequence; grey box, 2A peptide; 35S, 35S CaMV promoter; NOS, terminator sequence from *A. tumefaciens* nopaline synthase gene; (**c**,**d**) symptoms on systemically infected *N. benthamiana* leaves; (**e**) silver-stained SDS-PAGE of purified TBSV NPs: 1, 1 μg of E-coil-TBSV; 2, PageRuler Prestained Protein Ladder; 3, 1 μg of TBSV-wt. Black and grey arrows indicate TBSV-wt (41 kDa) and TBSV-E-coil (45 kDa) CPs, respectively; (**g**) silver-stained SDS-PAGE of purified PVX NPs: 1. 1 μg of E-coil-PVX; 2. 1 μg of PVX-wt; 3. Broad Range Prestained Protein Marker. Black and grey arrows indicate PVX-wt (25 kDa) and E-coil-PVX CPs (30 kDa), respectively. (**f**,**h**) Transmission Electron Microscopy images of TBSV-E-coil and E-coil-PVX purified NPs, respectively. Bars represent 200 nm.

**Figure 2 plants-13-00503-f002:**
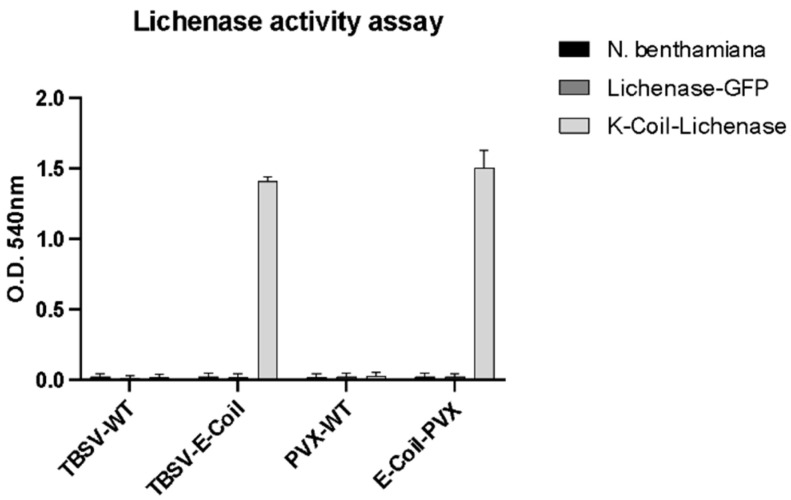
Lichenase activity assay. The X axis represents the different VNPs used for coating ELISA plates and tests the interaction and enzymatic activity with extracts derived from *N. benthamiana* negative control and Lichenase–GFP or K-coil–Lichenase agroinfiltrated plant extracts, as indicated. For coating, 1 μg of TBSV-wt or TBSV-E-coil NPs and 4 μg of PVX-wt or E-coil-PVX NPs were used. Data are presented as mean ± SD of biological triplicates.

**Figure 3 plants-13-00503-f003:**
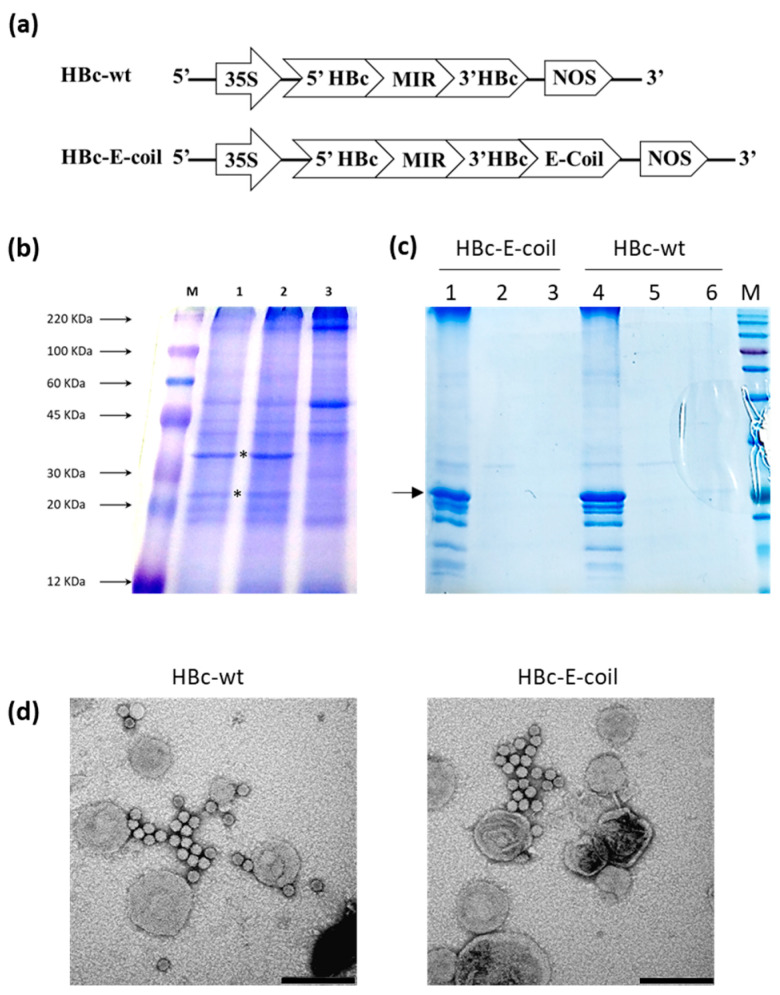
Production of HBc-wt and HBc-E-coil VLPs in plants. (**a**) Schematic diagram of genetic constructs: 35S, 35S CaMV promoter; NOS, terminator sequence from *A. tumefaciens* nopaline synthase gene; 5′ HBc: module encoding the 5′ region of HBc; MIR: module encoding the MIR region; 3′ HBc: module encoding the 3′ region of HBc; E-coil: module encoding for the E-coil peptide; (**b**) Coomassie-stained SDS-PAGE of total soluble proteins (TSP) extracted from plants agroinfiltrated with HBc-wt (1), HBc-E-coil (2), and p19 (negative control) (3) constructs; M, Color burst marker; 40 micrograms of TSP have been loaded for each sample; black asterisks indicate more intense differential bands compared to the control; (**c**) Coomassie-stained SDS-PAGE of purified VLPs: 1 and 4, a resuspended pellet of sucrose cushion; 2 and 5, sample loaded on the sucrose cushion and not entered in the cushion; 3 and 6, wash of the tube; M, Broad Range Prestained Protein Marker; arrow indicates the monomeric purified HBc subunit; (**d**) Transmission Electron Microscopy images of HBc-wt and HBc-E-coil purified VLPs. Bars represent 200 nm.

**Figure 4 plants-13-00503-f004:**
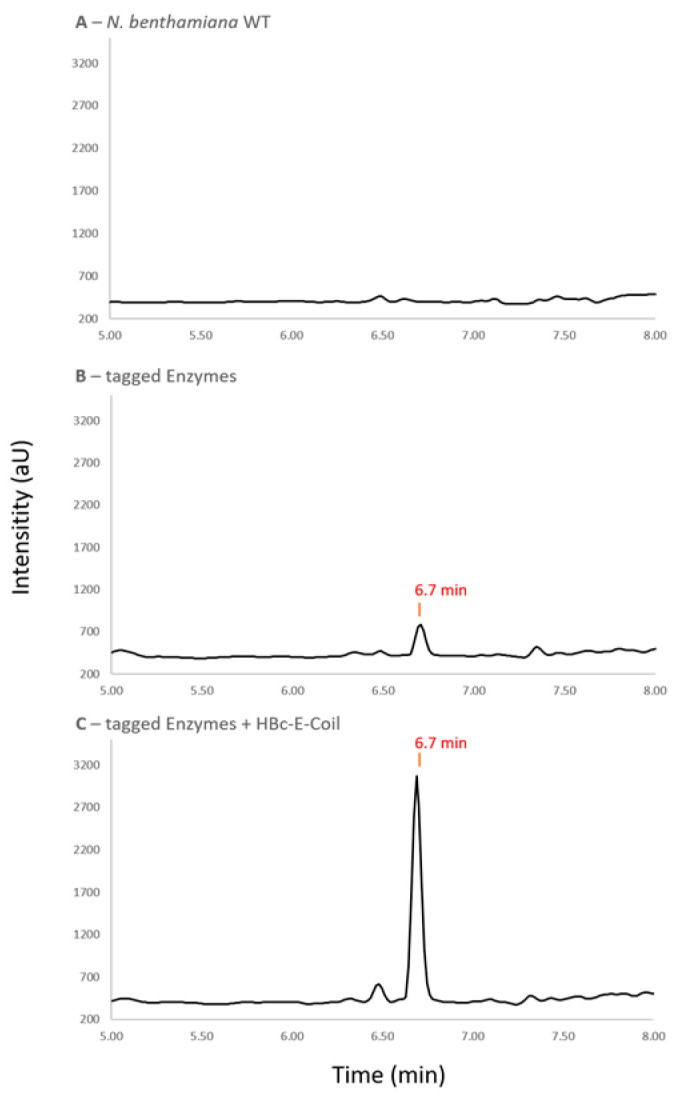
Metabolite analysis of plant extracts via HPLC-MS. Excerpt of chromatogram in negative single ion mode (385.2) for the detection of olivetolic acid glucoside (retention time = 6.7 min). (**A**) metabolite extract from *N. benthamiana* wild type; (**B**) plants expressing pathway enzymes tagged with K-coil (AAE1-K-coil, OLS-K-coil, and OAC-K-coil); (**C**) plants expressing pathway enzymes tagged with K-coil together with the E-coil functionalized HBc VLPs scaffold (HBc-E-coil).

**Figure 5 plants-13-00503-f005:**
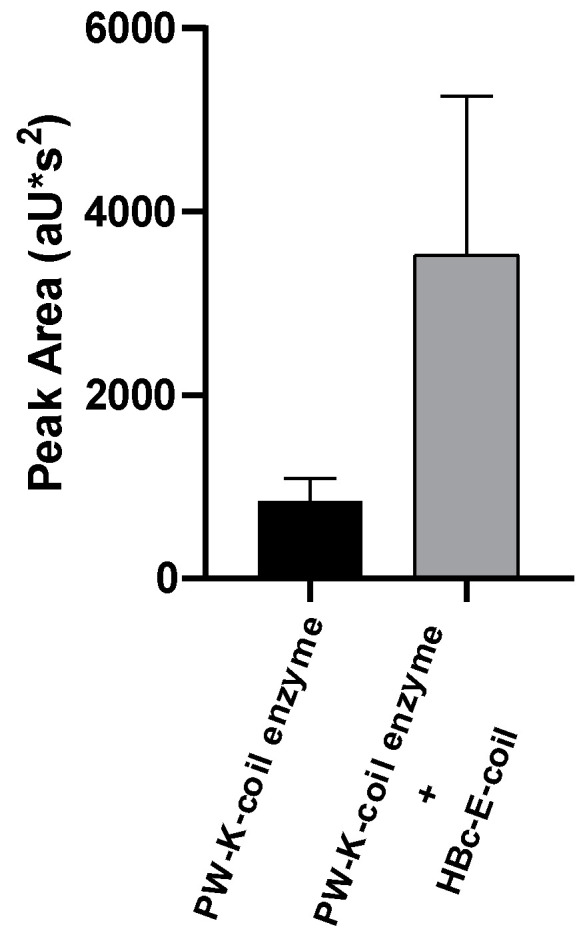
Detection of olivetolic acid glucoside formation in *N. benthamiana* leaves infiltrated with respective constructs. Statistically analyzed data represent mean of peak area underneath product peak. Error bars represent standard deviation (n = 30, *p* = 0.013 × 10^−9^).

**Table 1 plants-13-00503-t001:** Composition of the mobile phase for detection of olivetolic acid glycoside via HPLC. aqueous phase: H_2_O + 0.1% formic acid; organic phase: acetonitril +0.1% formic acid.

Time (min)	Aqueous Phase (%)	Organic Phase (%)	
0–2	80	20	gradient
2–8	0	100	isocratic
8–16	0	100	gradient
1–32	80	20	isocratic

## Data Availability

Data are contained within the article or Appendix A.

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
