# Peer review of "Plant-Produced Viral Nanoparticles as a Functionalized Catalytic Support for Metabolic Engineering"

_plants, 2024, doi:10.3390/plants13040503_

Round 1

Reviewer 1 Report

Comments and Suggestions for Authors

The title is very general, it does not reflect the objective of the study, the authors should consider giving a title that involves developing plant virus nanoparticles (VNPs) and plant-assembled virus particles (VLPs) as nano-scaffolds.

Introduction. Consider including recent articles on this research topic, as plant-based platforms with new viruses are being developed to provide an overview of these advances, by example: Arul SS, Balakrishnan B, Handanahal SS, Venkataraman S. Viral nanoparticles: Current advances in design and development. Biochimie. 2023 Aug 10; 219: 33-50. doi: 10.1016/j.biochi.2023.08.006.

Results and Discussion. Expand the discussion on Nicotiana benthamiana for the production of cannabinoids through intrinsic glycosylation of olivetolic acid, being a heterologous cannabinoid precursor biosynthetic pathway due to its importance in the present study, see: Gülck T, Booth JK, Carvalho Â, Khakimov B, Crocoll C, Motawia MS, Møller BL, Bohlmann J, Gallage NJ. Synthetic Biology of Cannabinoids and Cannabinoid Glucosides in Nicotiana benthamiana and Saccharomyces cerevisiae. J Nat Prod. 2020 Oct 23;83(10):2877-2893. doi: 10.1021/acs.jnatprod.0c00241

Update References

Author Response

Reviewer 1

The results of this study were not similar to other studies on possible applications of VNPs, but there are recent publications that should be included in the context of the potential of viruses to infect cells, and to be nanocarriers, having improved immunogenic properties, biocompatibility and biodegradability, see: Arul S.S. et al Viral nanoparticles: Current advances in design and development. Biochimie. 2023 Aug 10; 219: 33-50. doi: 10.1016/j.biochi.2023.08.006.

Answer: The reference and a sentence regarding other possible generic uses of VNPs have been added.

Change the title

Answer: In our view it would be misleading to change the title, i.e. adding the particular pathway used in the end. It is just an example and only providing a precursor for cannabinoid production. The majority of work lies on the decoration of various kinds of particles with active enzymes, as emphasiszed in the current title.

The conclusions are consistent with what was reported by Gülck T., et al., J Nat Prod. October 23, 2020;83(10):2877-2893; regarding the treatment of N. benthamiana for cannabinoid production that revealed intrinsic glycosylation of olivetolic acid.

Expand the discussion on Nicotiana benthamiana for the production of cannabinoids through intrinsic glycosylation of olivetolic acid, being a heterologous cannabinoid precursor biosynthetic pathway due to its importance in the present study, see: Gülck T, Booth JK, Carvalho Â, Khakimov B, Crocoll C, Motawia MS, Møller BL, Bohlmann J, Gallage NJ. Synthetic Biology of Cannabinoids and Cannabinoid Glucosides in Nicotiana benthamiana and Saccharomyces cerevisiae. J Nat Prod. 2020 Oct 23;83(10):2877-2893. doi: 10.1021/acs.jnatprod.0c00241

Answer: The introduction has been amended with the suggested topic and the additional reference has been included

Reviewer 2 Report

Comments and Suggestions for Authors

This manuscript is well written and very interesting. 

It may be helpful to define what the authors mean by E-coil in the Abstract. 

The concept and results shown are truly fascinating. 

A conclusion asection or at least concluding paragraph is needed, to summarize experiments and to describe future prospects for this technology, such as industrial applications. 

Author Response

We have added a conclusion section

Reviewer 3 Report

Comments and Suggestions for Authors

Sator and coauthors used viruses as a new alternative for producing biological nanomaterials that can be functionalized and act as enzymatic hubs. They proved that transient co-expression of the K-coil enzymes together with E-coil-VLPs allowed the establishment of the heterologous cannabinoid precursor biosynthetic pathway. My main comments are on the way the article is written and the presentation of the results. The introduction and results need serious editing and rewriting. Combining the Results section with the Discussion leads to the lack of commenting on the specific results.

The introduction must be rewritten and the references need to be included! Further, I am suggesting that the authors include a figure (in the Introduction) that can present their functionalized VLPs and their enzymatic activities.

L31 – L63 References are missing in this paragraph

L89-85 References are missing in this paragraph

L89 References

L108 References are missing in this paragraph

L95 Please include more detailed information about the heterodimerizing peptides as those derived from alpha-helical coiled-coil domains, such as glutamate coil (E-coil) and lysine coil (K-coil), and their functions.

In the introduction, I didn’t see any information about the lichenase. Please, include the information about the enzyme.

L104 Reference is missing

L112 Reference is missing

L114 While PVX and TBSV particles have been shown to be successfully decorated via K114 coil/E-coil interaction with an enzyme which retains its activity in an in vitro environment, a reference is missing here.

L116-119 Is the information from this paragraph becomes from your results? Rewrite the paragraph!

L120-122 A truncated form of the core antigen has been proven sufficient to form particles of 30 or 34 nm in diameter, depending on the number of dimers forming particles (90 or 120, respectively). References are missing!

More important, HBcAg does not need to incorporate nucleic acids for VNP. Please, include a reference!

Normally, HBcAg during its self-assembling incorporates heterologous nucleic acids. Please, comment on this!

L138-156 The information could be part of the Introduction, not from the results! Please, rewrite the paragraph!

Figure 1a and the different bars from the construct are not well explained!

L163 -165, L187 Please describe the individual parts of the structure in Figure 1b so that it is clear what is what!

Figure 1e and 1g, please indicate the molecular mass of the purified protein and molecular marker!

L162-168 Please, rewrite the sentence, it is too long and it is difficult to understand.

L173 What purification method of VLPs do you use? Please mention it here

How did you measure the concentration of the purified proteins?

L200-207 This information belongs to the introduction.

L259-271- This information belongs to the introduction.

For me, a discussion of the obtained results is missing, the results include paragraphs that look more like information from the introduction.

Author Response

Reviewer 3

The introduction must be rewritten, and the references need to be included! Further, I am suggesting that the authors include a figure (in the Introduction) that can present their functionalized VLPs and their enzymatic activities.

Answer: The introduction has been modified, implementing some paragraphs, and introducing new references where they were truly missing. Regarding the suggestion to introduce a figure in the introduction to represent the workflow of the experiments, we do not think it is useful to insert it here.

L89-85 References are missing in this paragraph

Answer: a reference has been added.

L89 References

Answer: it is the same lane indicated in the comment above, and now the reference has been added.

L108 References are missing in this paragraph

Answer: references are correctly inserted at the end of the paragraph (number 14 and 15). We moved reference 14 to anticipate the citation.

L95 Please include more detailed information about the heterodimerizing peptides as those derived from alpha-helical coiled-coil domains, such as glutamate coil (E-coil) and lysine coil (K-coil), and their functions.

Answer: We added more information in the text regarding heterodimerizing peptides in general, and regarding E-coil and K-coil peptides in particular. We also added 4 new references in support.

L104 Reference is missing

Answer: In this case we believe there has been a misunderstanding, since from line 104 (starting with “The aim of this work ……..”) we introduce the current work, the object, purpose, and approach of the study and the experimental design that will be detailed and discussed in the following sections. We consider this a common practice that can be found in the majority of published papers.

L112 Reference is missing

Answer: See previous comment

L114 While PVX and TBSV particles have been shown to be successfully decorated via K coil/E-coil interaction with an enzyme which retains its activity in an in vitro environment, a reference is missing here.

Answer: See previous comment

L120-122 A truncated form of the core antigen has been proven sufficient to form particles of 30 or 34 nm in diameter, depending on the number of dimers forming particles (90 or 120, respectively). References are missing!

Answer: references have been added

More important, HBcAg does not need to incorporate nucleic acids for VNP. Please, include a reference! Reference (16) is correctly present at the end of the paragraph, but we have added also another citation.

Normally, HBcAg during its self-assembling incorporates heterologous nucleic acids. Please, comment on this!

Answer: Indeed, it is worth noting that the incorporation of heterologous nucleic acids into HBcAg-based particles is a controlled and deliberate process found in the context of research and biotechnological applications, such as gene delivery. On the other hand, during a natural infection with HBV, the virus typically encapsidates its own genomic DNA rather than foreign nucleic acids.

However, in this context we exploited the outer surface of the virus for the attachment of enzymes to promote metabolic channeling. Even if nucleic acids of plant origin, such as messenger RNAs, would be entrapped inside the particles during assembly, which is indeed a likely possibility, it would not affect outer surface functionalization, being therefore irrelevant for the overall purpose of this approach.

Figure 1a and the different bars from the construct are not well explained! 

Answer: In each bar there is a letter whose meaning is correctly indicated in the figure legend. When the bar was too small to insert a letter, that box was indicated with a fill color, or lines, the meaning of which is clearly and correctly reported in the figure legend.

L163 -165, L187 Please describe the individual parts of the structure in Figure 1b so that it is clear what is what!

Answer: individual parts of the structure in Figure 1b are described in the figure legend, but we inserted details in the text too.

Figure 1e and 1g, please indicate the molecular mass of the purified protein!

Answer: details have been added both in the text and in the figure legend.

L162-168 Please, rewrite the sentence, it is too long and it is difficult to understand.

Answer: this part has been rephrased.

L173 What purification method of VLPs do you use? Please mention it here

Answer: purification protocols, different for the two viruses but both based mainly on ultracentrifugation,

are well-consolidated by previous works of the group, and to shorten the materials and methods section it was preferred to cite them (references 14, 39 and 40). We now added some little hint here.  

How did you measure the concentration of the purified proteins?

Answer: Quantification method is explained in the Materials and Methods section. We now added here some little hint.   

L31 – L63 References are missing in this paragraph

Answer: To our understanding this is the abstract which should not contain references

In the introduction, I didn’t see any information about the lichenase. Please, include the information about the enzyme.

Answer: Information has been included in the introduction

L116-119 Is the information from this paragraph becomes from your results? Rewrite the paragraph!

Answer: The paragraph has been rephrased., although this information in our view is necessary to understand the various systems being used.

L138-156 The information could be part of the Introduction, not from the results! Please, rewrite the paragraph!

Answer: In our view the introduction gives an general state-of -the-art view and background. Nevertheless it helps the reading flow when reasons for choosing certain experiments is addressed in the results section as well. This improves clarity.

L200-207 This information belongs to the introduction.

Answer: The introduction has been modified accordingly

L259-271- This information belongs to the introduction.

Answer: The introduction has been modified accordingly

For me, a discussion of the obtained results is missing, the results include paragraphs that look more like information from the introduction.

Answer: We combined results and discussion to enable the instantaneously discussion due to the manifold aspects of the project.

Additionally, we included a conclusion in the end

Round 2

Reviewer 3 Report

Comments and Suggestions for Authors

I am satisfied with the writers’ answers and I agree that a manuscript can be published in this format.